# Gut-on-a-Chip Models: Current and Future Perspectives for Host–Microbial Interactions Research

**DOI:** 10.3390/biomedicines11020619

**Published:** 2023-02-18

**Authors:** Moran Morelli, Dorota Kurek, Chee Ping Ng, Karla Queiroz

**Affiliations:** MIMETAS B.V., 2342 DH Oegstgeest, The Netherlands

**Keywords:** organ-on-a-chip, gut-on-a-chip, microfluidics, host–microbial interactions, organoids, drug screening, microbiome, disease modeling

## Abstract

The intestine contains the largest microbial community in the human body, the gut microbiome. Increasing evidence suggests that it plays a crucial role in maintaining overall health. However, while many studies have found a correlation between certain diseases and changes in the microbiome, the impact of different microbial compositions on the gut and the mechanisms by which they contribute to disease are not well understood. Traditional pre-clinical models, such as cell culture or animal models, are limited in their ability to mimic the complexity of human physiology. New mechanistic models, such as organ-on-a-chip, are being developed to address this issue. These models provide a more accurate representation of human physiology and could help bridge the gap between clinical and pre-clinical studies. Gut-on-chip models allow researchers to better understand the underlying mechanisms of disease and the effect of different microbial compositions on the gut. They can help to move the field from correlation to causation and accelerate the development of new treatments for diseases associated with changes in the gut microbiome. This review will discuss current and future perspectives of gut-on-chip models to study host-microbial interactions.

## 1. Introduction

The intestine plays a pivotal role in health and disease. Its principal function is to absorb nutrients from the food we ingest, but it also participates in drug transport, metabolism, and the secretion of essential hormones. Furthermore, the gut wall is home to large microbial communities, which participate in homeostasis through protection against pathogens [1,2] and the production of vitamins and short-chain fatty acids [3]. Gut dysbiosis—a disruption of the intestinal microbiota—has been associated with several pathologies, such as inflammatory bowel disease (IBD), cancer, obesity, and diabetes [4,5,6]. Although many models have been developed to study intestinal functions in health and disease, translation to human in vivo settings remains challenging. This is mainly because existing models fail to fully recapitulate the complex multi-component composition of the intestinal mucosa. For example, most in vitro models are cultured in static conditions, where the co-culture with living microbes is only feasible for a limited period because of rapid microbial overgrowth [7]. Moreover, investigating the interactions between gut microbes, which are often obligate or facultative anaerobes, and oxygen-requiring epithelial cells is technically challenging [8,9,10]. Over the last few years, more sophisticated models have been developed, including organoids and gut-on-a-chip. These models can revolutionize our understanding of the gut microbiome and its role in health and disease and pave the way for developing new therapeutic strategies. 

There have been several reviews on related topics, such as the evolution of intestinal models from cell lines to gut-on-chips [10], the use of organ-on-chip to model the intestine [11], microbial ecosystem modeling [12], and the use of organoids and organ-on-chip models to study host–microbial interactions [13]. However, to our knowledge, there has not been a recent comprehensive review that provides a list of studies using organ-on-chip to study host–microbial interactions. Here, we gathered publications on the co-culturing of host and microbial cells in microfluidic platforms and discussed the main findings and the key parameters to consider in developing such models (Figure 1). We also examined the advantages, limitations, and future potential of these models to better understand intestinal physiology.

## 2. Gut-On-Chip Systems

### 2.1. Material

Polydimethylsiloxane (PDMS) is the most widely used material to produce microfluidic systems [14]. Its ease of production and ability to be molded into different shapes revolutionized microfluidics and made it accessible to most labs. PDMS is an elastomeric polymer with various advantageous traits for biomedical use, such as biocompatibility, gas permeability, optical transparency, chemical stability, and tunable elasticity and wettability [14,15].

Despite these advantages, PDMS has some limitations. It can absorb small hydrophobic molecules from- or release free oligomers into- the culture medium, which can affect experimental outcomes [14,15,16]. Mathematical models have been developed to simulate drug concentrations in PDMS microfluidic chips [17]. Several coating methods have emerged to overcome small molecule partitioning into PDMS, but studies showed that absorption was variable, time-dependent, and not determined exclusively by hydrophobicity [18,19]. While such methods can mitigate the absorption of hydrophobic molecules and the release of free oligomers into the culture medium, they often slow the fabrication process and diminish elasticity, transparency, and biocompatibility [15].

Although PDMS has led to a wide range of developments for biomedical applications, further work is still needed to improve current surface treatments and the scalability of the production of PDMS-based devices [14]. These points will have to be addressed to transfer microfluidic technology from laboratories to industry and, therefore, to the biotechnology market. For this reason, more and more researchers are turning to PDMS-free alternatives for the construction of organ-on-a-chip models [20,21,22,23,24].

### 2.2. Flow

Flow is an inherent property of microfluidic gut-on-chip models, where perfusion of the cell culture medium at defined flow rates is achieved through a variety of methods such as syringes [25,26,27], pumps [21,22,28,29] or gravity [23,30,31,32]. The perfusion of cells can replicate in vivo fluid flow, exposing cells to the corresponding shear stresses and increasing physiological relevance.

Several examples of improved cell differentiation have been reported in different microfluidic devices [33,34,35]. When comparing Caco-2 cells cultured in static or dynamic conditions, fluidic culture accelerated the formation of leak-tight gut tubules [23], improved morphogenesis [20,25,35], and enhanced barrier integrity [20,21,23,36]. It also increased CYP3A4 activity [31,35], glucose absorption [37], and mucus secretion [36]. Kulthong et al. showed that Caco-2 cultured in static or dynamic conditions showed different responses to nanomaterials [38]. Workman and colleagues found that the culture of IPSCs-organoids under a continuous flow resulted in the formation of a polarized epithelium containing all differentiated subtypes and stem cells [34].

At the transcriptome level, intestinal cells showed different profiles in static or dynamic conditions [39]. Kasendra et al. showed that the transcriptome of duodenal organoids, when cultured in a microfluidic device, more closely mimicked human duodenum compared to the organoids used to prepare the chips [40,41].

Flow also prolongs epithelial integrity by removing dead cells and improving access to nutrients [42,43]. This is especially useful in models of host-microbial interactions, where microbial overgrowth is an important limitation [44]. Using a microfluidic device with the flow and cyclic strain, Kim and coworkers co-cultured Caco-2 and *L. rhamnosus* GG (LGG) for up to a week without impairing barrier integrity. In contrast, loss of barrier function and cell death was observed after only 48 h in a transwell system [25].

While it is now clear that fluid flow and corresponding shear stresses can improve physiological relevance, only a few studies have explored which flow profiles are best for intestinal epithelial cell differentiation. Using a two-channel system, Shin et al. showed that basolateral flow was crucial for epithelial morphogenesis [45]. Langerak and colleagues showed that pump-driven flow improved cell differentiation compared to gravity-driven flow [46]. Fois et al. compared a dynamic flow rate to a constant flow rate and showed that the dynamic flow rate had no biological benefit but a reduced reagent consumption, which is an essential parameter for scalability [47]. Further work will have to investigate the effect of flow parameters such as shear stress, dynamic or constant flow, and—in the case of multicompartment platforms—apical and basolateral flow profiles.

### 2.3. Mechanical Forces

The intestine undergoes ring and segmental contractions, which serve a propulsive function. Their frequency range from 2 to 20 per minute, with high frequencies towards the duodenum and low frequencies towards the colon [48,49,50,51]. Increasing evidence suggests that these mechanical forces also influence homeostasis and intestinal development. For example, alterations in gut motility play a role in inflammation and several disorders, such as inflammatory bowel disease (IBD) [48,52]. Moreover, several studies showed that intestinal mobility and gut microbiota were inter-regulated [8,53,54,55,56].

A limited number of studies have explored the impact of mechanical forces on intestinal epithelial cells (IECs) development, homeostasis, and interaction with microbes. Kim et al. used a system of vacuum chambers and cyclic suction to mechanically deform Caco-2 and mimic peristalsis. Cyclic strain did not affect cell morphology nor trans-epithelial electrical resistance (TEER) values but increased paracellular permeability and aminopeptidase activity [25]. Later, they observed that the cessation of cyclic strain—but not flow—was sufficient to induce the overgrowth of GFP-labelled *E. coli* on Caco-2 cells [57]. Using a similar device, Grassart and coworkers observed that cyclic strain enhanced *Shigella* invasion in Caco-2 cells, indicating that the bacteria took advantage of the gut micro-architecture to increase its virulence [58]. In a similar study, Bouquet-Pujadas et al. studied the effect of peristalsis on the invasion of two enteroinvasive microbes: *S. flexneri* and *E. histolytica*. They observed that the virulence of both pathogens was increased by peristalsis, even though the invasion mechanisms of each pathogen were different [59].

In jejunal organoids, cyclic stretch did not affect the response to a bacterial virulence factor [60]. In colon organoids cultured in a microfluidic device, stretching did not have a clear impact on the polarization of IECs but enhanced the terms relating to the transport of water, ions, and lipoproteins at the transcriptome level [61]. Adding mechanical stretch on top of flow did not contribute to the further differentiation of jejunal organoids [33].

Jing and colleagues used a 3-channel system and a multi-channel pneumatic pump to regulate air pressure to the fluid medium, allowing for the generation of a periodic and physiologically relevant pressure difference between the middle channel and the two adjacent channels. They observed that periodic peristalsis promoted the growth and differentiation of epithelial cells compared to static transwell models [28]. However, because this study compared a microfluidic device with flow and peristalsis to a static transwell model, it was impossible to assess whether the results obtained were a direct consequence of peristalsis alone.

A recent computational study showed that, together with pore size, peristalsis was one of the critical factors affecting shear stress on the membrane surface of gut-on-chip systems, indicating that these factors could play a pivotal role in cell differentiation [62]. Although increasing evidence shows that mechanical forces influence the biology of intestinal mucosal cells [48,63], not many models have explored this perspective. To this day, the standard method to include peristalsis is the use of a cyclic strain through vacuum chambers [25], which was initially designed to mimic breathing in lung models and might, therefore, not fully mimic peristalsis [33]. The effect of different types of mechanical stimuli on intestinal cultures will have to be investigated to determine whether current approaches faithfully replicate in vivo situations.

### 2.4. Oxygen-Gradient

A descending oxygen gradient is present along the length of the intestine and across its wall. In the small and large intestines, the lumen is in a state of physiological hypoxia and is home to trillions of microbes, most of which are strict anaerobes. Although there is a strong need to develop more representative models of the intestine, mimicking physiologic intestinal hypoxia in vitro remains challenging because human cells require oxygen while standard cell culture methods are performed in an aerobic environment.

The first microfluidic model allowing the co-culture of human cells and anaerobic microbes in separated compartments was reported in 2016 and called human-microbial cross-talk (HuMiX). Shah et al. successfully co-cultured Caco-2 with *L. rhamnosus* or the strict anaerobe *B. caccae* for 48 h and showed a differential response between the two species. Oxygen sensing was performed using 5 mm-wide optical sensors [21].

Furthermore, Shin and colleagues developed the anoxic–oxic interface-on-a-chip using an anoxic medium to generate a controlled oxygen gradient, which allowed the culture of Caco-2 to be in direct contact with two obligate anaerobes for up to a week [64]. They performed in situ measurements using dendrimer-encapsulated nanoparticles (Pt-DENs) [65]. In a later iteration, they developed the physiodynamic mucosal interface-on-a-chip (PMI chip), containing a convoluted microchannel, in which they co-cultured Caco-2 with human fecal microbiome [66].

In addition, Jalili-Firoozinezhad et al. incorporated physiologic oxygen gradients into a gut-on-a-chip and co-cultured Caco-2 or primary ileal organoids in direct contact with complex fecal-derived microbiota for five days. The authors measured real-time oxygen concentration using fluorescent sensors [67].

Finally, De Gregorio and coworkers reproduced the architecture and vertical topography of the microbiota in an immune-competent gut-microbiota axis model (MihI-oC) with a complex serosal environment composed of a responsive extracellular matrix (ECM) and the release of immune mediators from various cell types (epithelial, stromal, blood, microbes). The authors measured oxygen using an optical detector (OXY-4 PreSens) [68].

One of the main limitations of previously described anaerobic gut-on-chip models is that none showed whether intestinal epithelial cells were in normoxic conditions. To address this issue, Wang et al. described a standalone microfluidic device with fluorescent oxygen sensors and partitioned the oxygen environment to allow cells to remain under normoxic conditions while bacteria were under anoxic conditions. To verify the oxygen status, the authors used three complementary methods––(i) an in-situ oxygen sensor was used to measure oxygen levels in the device, (ii) pimonidazole was used to measure oxygen levels of the intestinal cells, and (iii) a GFP facultative anaerobe was used to measure oxygen levels of the bacteria [69]. The combination of these methods enabled a more comprehensive monitoring of oxygen concentrations within the device and allowed for the regulation of oxygen tension to meet the needs of both host cells and anaerobic bacteria; this is an improvement upon the monitoring methods described in previous reports.

In the last five years, the emergence of new models recapitulating physiologic hypoxia has allowed the co-culture of strict anaerobes with human cells. The improvement and miniaturization of oxygen sensors enabled the development of new platforms with integrated oxygen sensors and higher throughput [22,70]. Microfluidic models play a pivotal role in improving the simulation of the intestinal oxygen gradient and recapitulating physiological conditions to co-culture aerobic and anaerobic micro-organisms and better mimic host-microbial interactions.

### 2.5. Micro-Architecture

When viewed from inside the lumen, the epithelium of the colon appears flat, while the epithelium of the small intestine displays finger-like projections referred to as villi. Both the small intestine and colon have invaginations called crypts, which contain stem cells at the bottom and differentiated cells at the top [71]. Several microfluidic models were used to study the spatial arrangement of intestinal cells and their role in homeostasis and host-microbial interactions.

Kim et al. showed that Caco-2 spontaneously formed villi-like undulating structures when exposed to a peristalsis-like motion and fluid flow. Such arrangements were similar to the microenvironment of the intestine, with proliferative niches located in the crypts and differentiating cells moving along the crypt-villi axis to differentiate into all subtypes of intestinal epithelial cells [35]. They later showed that microbial cells predominantly colonize intervillous spaces [57]. Using a similar model Grassart and colleagues showed that crypt-like structures were critical for Shigella adhesion [58]. Similar crypt-villi-like structures were observed in other gut-on-a-chip models using Caco-2 [20], duodenal organoids [40,41], and IPSCs organoids [34], but not in jejunal organoids [33].

Other groups used microfabricated scaffolds to mimic intestinal micro-architecture. Wang et al. showed that chemical gradients applied to an epithelial monolayer and cultured on a microfabricated scaffold recapitulate in vivo responses of intestinal crypts [72]. Nikolaev et al. used laser-cut scaffolds to develop a gut-on-a-chip model to re-create the spatial arrangement of the crypt-villus axis, which remained stable for several days in culture and allowed long-term host–microbial studies [42]. Shin and colleagues co-cultured fecal microbiota and Caco-2 cells in a convoluted channel with multiaxial deformations, which induced dynamic cell trains and enhanced luminal particle mixing [66]. Using a collagen-scaffold mimicking intestinal villus, Shim et al. observed improved metabolic activity in cells cultured in 3D on the scaffold, compared to cells in 2D. However, they also observed a higher permeability and lower villi height, indicating that the potential increased shear stress at the top of the villi could be detrimental to the cells [31]. By pouring collagen into a PDMS mold, Verhulsel and coworkers developed a model with the topography and dimensions of the mouse gut, in which they co-cultured primary mouse epithelial cells and fibroblasts in conditions closely resembling the in vivo intestinal epithelium [43].

Two-dimensional models are incapable of fully recreating the micro-architecture of tissues. With the advent of 3D cell culture models, it became possible to replicate the micro-architecture of the intestine and examine the spatial arrangement of cells in homeostasis or disease. Further development in microfabrication techniques, such as 3D printing, would allow the combination of micro-scaffold with extracellular matrices to better mimic the gut microenvironment and create more relevant intestinal models.

## 3. Host Component

### 3.1. Intestinal Epithelial Cells

#### 3.1.1. Epithelial Cell Lines

The colonic adenocarcinoma-derived cell line Caco-2 is frequently used in vitro models of the intestinal mucosa. Caco-2 spontaneously polarize and differentiate into a heterogenous population of intestinal epithelial cells, expressing tight junctions, microvilli, and several enzymes and transporters [73].

Caco-2 are typically cultured on transwell inserts, where their polarized orientation allows the separation of the apical and basolateral sides for transport studies and the evaluation of barrier integrity by TEER. The Caco-2 transwell model is also used for host–microbial interactions [10], but its static nature makes it difficult to establish long-term stable co-cultures because of rapid microbial overgrowth [74].

In recent years, the emergence of microfluidic gut-on-chip devices has allowed us to better mimic the intestinal environment. Microfluidic flow provides a constant supply of nutrients and enables waste removal and better cell differentiation [23]. Several gut-on-chip models co-cultured Caco-2 cells and microbes to study the intestinal barrier function and host–microbial interactions. The first gut-on-a-chip model co-culturing human cells and living microbes was described in 2012 by Kim et al. They co-cultured Caco-2 cells with the commensal microbe LGG and observed an improved barrier function similar to that observed in humans [25]. This model was also used to study shigella infection [58] and SBV virus infection [75].

Other common intestinal cell lines include HT-29 and T84. HT-29 cells are derived from human colorectal adenocarcinoma [76], and their mucous-secreting subclone HT-29-MTX can be used in combination with Caco-2 to study intestinal inflammation [77,78,79]. T84 cells are derived from the lung metastasis of colon carcinoma [80,81] and can be used to model colonic epithelium [82].

Despite the many advantages of cell lines, i.e., cost-efficiency, robustness, high throughput, and ease of use, there are limitations [83]. For example, when compared to human biopsies, Caco-2 cells have a different expression of tight junction proteins, enzymes, and transporters [84]. Caco-2 cells do not produce mucus, which is essential in the adhesion and invasion of pathogens [85]. HT-29 form a leaky barrier and have an impaired glucose metabolism [10]. Lastly, because of the heterogeneity of tumor-derived cell lines, diverse culture conditions can result in the selection of subclones, resulting in experimental variability across different experiments or laboratories [86,87].

#### 3.1.2. Organoids

Organoids are self-organized 3D microtissues cultured in an extracellular matrix and can be derived from patient biopsies [88] or induced-pluripotent stem cells [89]. They are enclosed structures containing multiple organ-specific cell types that are grouped and arranged in a similar way as an organ. There is now extensive data showing that organoids can recapitulate some features of their corresponding organ, opening new perspectives in disease modeling and personalized medicine and potentially replacing animal experiments.

Organoids have emerged as a promising tool to model intestinal physiology and host–microbial interactions because they contain numerous specific cell subtypes and can recreate regional identity [90]. Additionally, patient-derived organoids can show a disease phenotype when isolated from diseased tissue. An example is organoids derived from colorectal cancer patients, which show gene expression signatures similar to cancer cells in vivo [91,92]. Similar results were observed in organoids from patients with ulcerative colitis [93,94] and Crohn’s disease [93,95].

Intestinal organoids are also being used to study host–microbial interactions [13,96,97]. In most cases, microbes are cultured in suspension [98,99] or are microinjected into the organoid lumen [100,101]. Several pathogens, such as *V. cholerae* [102], *S. flexneri* [103,104], enterohemorrhagic *E. coli* [105,106], and *H. pylori* [100], have been studied in organoid models. The interactions between the intestinal mucosa and commensal microbes, such as *lactobacilli,* have also been studied in organoids [107].

Nevertheless, there are several limitations associated with the use of enclosed matrix-embedded organoids. Access to the apical side of the epithelium is difficult, which limits sampling, transport studies, drug exposure, or co-culture with microbes. Micro-injection techniques have been developed to overcome this issue, but they require specific knowledge and equipment [108,109]. Another limitation is that the evaluation of barrier integrity is difficult, while loss of barrier integrity is a key hallmark of many intestinal diseases [110].

Another way of accessing the luminal side of organoids is to dissociate and seed them in transwells. While grown on inserts, organoids form a polarized monolayer with easy access to both the apical and basolateral compartments and allow barrier evaluation through TEER measurements, contrary to matrix-embedded organoids [111]. Roodsant et al. used small intestinal organoids cultured as a monolayer on inserts to study infection with *Enterovirus A71* and *L. monocytogenes*. They observed viral replication and bacterial translocation and were able to monitor the subsequent pro-inflammatory host response [112]. Angus and colleagues developed an autologous colonic monolayer model using patient-derived material from IBD or non-IBD patients. They observed that epithelial integrity was compromised in monolayers from IBD patients and further impaired when co-cultured with bacteria [113].

One of the most promising applications of organoids is to dissociate and culture them under perfusion in microfluidic devices to generate more physiologically relevant models. Several gut-on-a-chip models using organoids have been described, including IPSCs-derived organoids [34,114], or patient-derived organoids from the human duodenum [40,41], jejunum [33,60], ileum [30,67], colon [29,115,116], or rectum [115]. Some of these models have been used to study host–microbial interactions. Tovaglieri et al. used colon organoids in a gut-on-a-chip system to recapitulate species-specific differences in the tolerance to enterohemorrhagic Escherichia coli (EHEC) infection. They showed that EHEC-induced epithelial injury was higher in humans than mice and discovered four human microbiome metabolites, which induced flagellin expression and were responsible for this species-specific difference [29]. Using a mouse organoid model, Gazzaniga and colleagues studied *Salmonella typhimurium* infection and could replicate epithelial injury. They colonized their chip with a complex mouse microbiota along with *S.typhimurium* and used 16s sequencing to identify *Enterococcus faecium* to be protective against *S.typhimurium* [117].

In summary, organoid technology provides a powerful tool for studying intestinal physiology and pathogenesis. Yet, many challenges still need to be addressed, such as the scalability or the co-culture with other cell types to fully recapitulate the physiological microenvironment. Also, while patient-derived biopsies can be used for precision medicine, their availability remains limited. For organoids to be used routinely to determine patient responses, these need to be established from different sites and recreate the features of their original tissue. Moreover, they need to be compliant with freeze/thaw procedures to test drugs at different time frames. The establishment of organoid biobanks and the standardization of organoid generation and culture represent fundamental steps for further pre-clinical research [91,118,119,120].

### 3.2. Mucus

The intestinal mucus forms a barrier between the epithelium and the lumen. It is composed of two layers: a dense inner layer impermeable to microbes and a soft outer layer, which is home to the commensal flora. The loss of mucus homeostasis, such as changes in the mucus layer, abnormal post-translational modifications, and alterations in the expression of key mucins, are important factors in the pathogenicity and severity of several diseases, including IBD [121].

Despite its importance, not many organ-on-a-chip models include a mucus component. Hagiwara et al. applied a mucus layer on the surface of Caco-2 cells to protect them from bile acids and simulate intestinal fluid [122]. Sontheimer and coworkers used patient-derived colon organoids to develop a model containing mucus-secreting goblet cells forming a mucus bilayer with a total thickness similar to observations in vivo. The authors used live, noninvasive visual analysis to track the accumulation of mucus over time and observed that the thickness of the mucus layer increased after exposure to the pro-inflammatory mediator PGE2. Further analysis revealed that this increase was not due to the secretion of new mucus but rather to changes in the hydration state of the pre-existing mucus and mediated by ion secretion through NKCC1. Notably, the inner layer of the mucus was preserved during the PGE2-induced swelling of the outer layer, suggesting that despite the changes in its hydration state, the mucus layer maintained its structural integrity [116].

### 3.3. Immune Cells

Various studies used organ-on-a-chip platforms to investigate the role of immune cells in intestinal physiology and pathology. Using the membrane-free OrganoPlate model [23], Gijzen et al. assembled four cell lines, including the immune cells THP-1 and MUTZ-3, into an immunocompetent platform to study intestinal inflammation [78]. Using the same platform, Beaurivage and colleagues combined patient-derived colon organoids and monocyte-derived macrophages embedded in ECM to model inflammatory processes [115]. In a similar approach, neutrophils were incorporated into the vascular compartment of OrganoPlate to study immune migration. In inflammatory conditions, neutrophils could migrate through the extracellular matrix from the vascular to the intestinal compartment [123].

Kim et al. co-cultured pathogenic or non-pathogenic strains of *E. coli* with intestinal epithelial cells and peripheral blood mononuclear cells (PBMCs) to build an IBD model. They showed that immune cells must be present with lipopolysaccharide (LPS) or pathogenic bacteria to cause villus injury and the loss of barrier integrity. Moreover, probiotic and antibiotic therapies could suppress villus injury caused by pathogenic bacteria [124]. A similar model showed that reduced barrier integrity, as observed in the leaky gut syndrome, led to higher susceptibility to microbial infections and higher inflammation [125].

Maurer and colleagues developed an immunocompetent system, including PBMC-derived dendritic cells and macrophages, Caco-2 cells, and HUVECs. The model showed physiological immune tolerance in the intestinal lumen, where pre-colonization by *L. rhamnosus* reduced *C.albicans*-induced tissue damage. They used the MOtiF platform, a device made of polystyrol (PS) and consisting of two channels separated by a porous polyethylene terephthalate (PET) membrane [20].

Jing et al. assembled Caco-2, primary human macrophages, and HUVECS in a microfluidic chip to study enteritis and its modulation by chitosan oligosaccharides (COS). They stimulated the system with dextran sodium sulfate (DSS) or pathogenic *E. coli* and observed that COS could protect both the intestinal and vascular barriers by preventing the attachment and infiltration of pathogenic *E. coli* [126].

### 3.4. Vascular Cells

Several gut-on-a-chip models have included a vascular component to re-create the interface between intestinal and endothelial cells [11,20,28,29,30,36,40,41,67,126,127]. In most models, epithelial cells were seeded on one side of a membrane, while an endothelial monolayer was seeded on the opposite side.

Kasendra et al. showed that adding human intestinal microvascular endothelial cells (HIMECs) in their gut-on-a-chip system accelerated the differentiation of duodenum organoids and improved barrier integrity [40]. Using a different system, Jing and colleagues showed that the presence of vascular endothelial cells (HUVECs) improved aminopeptidase activity and the morphology of Caco-2 cells [28]. Jeon et al. used a membrane-free system containing three compartments to culture Caco-2 cells with HUVECs. The epithelial cells were seeded in the left channel, a collagen-I gel was seeded in the middle channel, and HUVECs were added to the right channel. The co-culture of epithelial and vascular cells resulted in the expression of polarized and differentiated columnar epithelium, which was not observed in Caco-2 monoculture [36].

Although many gut-on-a-chip models included vascular cells, no study so far has investigated the interaction between epithelial and vascular compartments. This means that the observed differences, such as improved barrier integrity or diverse transcriptomic profile, cannot be attributed to the endothelial cells, as they could also come from the medium used to culture these cells. Further work will have to be conducted to characterize this type of co-culture and explain the results obtained.

### 3.5. Fibroblasts

Although it is now clear that fibroblasts regulate the intestinal epithelium in health and disease [128,129], most existing gut models lack the stromal compartment or contain an exogenous extracellular matrix (ECM). The first gut-on-a-chip model, including a stromal component, was published in 2020 by Seiler and colleagues. They constructed a microfluidic model and co-cultured patient-derived subepithelial myofibroblasts (ISEMFs) with endothelial cells (ECs). They showed that ISEMFs have angiogenic properties in response to interstitial pressure generated by microfluidic culture. They then included patient-derived intestinal epithelial cells on a porous membrane on top of the perfused vasculature composed of ECs and ISEMFs [30]. Verhusel and coworkers used organoid-derived mouse epithelium cultured on a 3D collagen scaffold containing primary mouse intestinal fibroblasts to generate a model with cell morphology resembling the in vivo epithelium [43].

De Gregorio and colleagues recently proposed the Microbiota-Intestine axis on-chip (MihI-oC) consisting of a responsive ECM and various cell types such as epithelial, stromal, blood, and microbial species under homeostatic or inflamed conditions. Upon stimulation with LPS, the study showed the protective role of microbiota on the barrier function and stromal compartment, while a lack of microbiota resulted in an altered collagen fiber assembly and increased ROS production. Additionally, the presence of microbiota impacted cytokine secretion at the luminal and basolateral sides [68].

## 4. Microbial Component

Several approaches have been used to include a microbial component in gut-on-a-chip models. Most studies used single strains of pathogens [29,58,59,126], probiotics [21,25,36], or both in combination [20,27,28,57,125,130]. Some studies also used VSL#3: a defined microbial community of probiotics [57,125,131]. While most models co-cultured human and microbial cells in aerobic conditions, several platforms included an anaerobic compartment with strict anaerobes [21,64,68] or complex microbiota derived from fecal samples [66,67,117]. Gut-on-chip models were also used to study viruses [75,132], parasites [42], phages [133], live-biotherapeutic products (LBP) [127,131], or microbial toxins [60].

Although most studies focused on developing and characterizing gut-on-chip models to co-culture host and microbial cells, several groups showed potential clinical applications. Nelson and colleagues used a gut-on-chip system to characterize a live biotherapeutic product to treat phenylketonuria [127]. Min et al. used probiotics in a leaky gut-on-chip and observed an amelioration of the barrier integrity and reduction in inflammation [131]. Zhao and colleagues observed that the antibiotic Amikacin could efficiently inhibit bacteria-induced inflammation, as observed in clinical studies [27]. These examples show how gut-on-chip models could be used to test new treatments, such as live biotherapeutic products or antibiotics. We expect more studies showing clinical applications of gut-on-chip models to be released in the next few years.

Thus, gut-on-chip models are promising for studying host-microbial interactions, but many challenges still need to be addressed. Microbial infections in gut-on-chips are typically performed over days to weeks and, therefore, do not recapitulate the long-term relationships between host cells and microbes [101]. Moreover, to transition from single microbial species to complex communities, most studies use fecal samples, which do not fully represent the mucosal microbiota [134]. Alternatives such as mucosal-luminal interface aspirates could complement stool samples to better replicate the regional diversities of the gastrointestinal microbiome [134,135]. Another solution could be the combination of gut-on-chip with bioreactors. Microbes or microbial metabolites could be obtained after long-term culture from conditions mimicking different parts of the digestive tract in a healthy or diseased state [12,136].

In summary, despite varying levels of complexity among the current models, none can fully replicate all the crucial elements present in the human gut [12]. The potential of organ-on-a-chip to model long-term host–microbial interactions using region-specific human cells and microbial communities from healthy and diseased individuals still needs to be demonstrated. Nevertheless, organoids and organ-on-a-chip are set to play an essential role in the development of new in vitro models to better understand host–microbial interactions.

## 5. Future Directions

Over the last ten years, microfluidic systems have emerged as a novel approach to developing new models with increased physiological relevance. Yet the most described gut-on-chips have limitations such as high costs, low throughput, the requirement of specialized equipment, or lack of compatibility with important readouts.

In 2017, Trietsch et al. described the OrganoPlate, a membrane-free microfluidic platform that allows the culture of perfused gut tubules and the rapid assessment of their barrier integrity [23,78,114,115,137]. The OrganoPlate consists of 40–96 microfluidic chips patterned underneath a 384-well micro-titer plate and is compatible with standard microscopes or robots as well as high-throughput assays such as reactive oxygen species (ROS) quantification [138] or TEER measurement [139]. Variations of this platform have been used for immune cell migration [123,140,141], high-throughput compound toxicity screening [142], and the in vitro grafting of spheroids and organoids on a microfluidic vascular bed [143]. Recently, the effects of the microbial toxin deoxynivalenol (DON) were tested using OrganoPlate. The authors examined the dose and route of exposure and found that barrier impairment occurred at higher concentrations than in static models. They also observed that the barrier was more sensitive when the toxin was added to the basolateral side, although DON had to pass the ECM [144]. This supports the idea that gut-on-chip models with a higher throughput could be used to assess the effects of microbial toxins and identify compounds with modulatory properties.

Other high-throughput organ-on-a-chip platforms have been described. Azizgolshani et al. reported on a system that was compatible with several readouts. The platform comprises 96 chips composed of two channels with programmable flow control and integrated sensing for TEER and oxygen [22]. The platform can re-create physiologic flow profiles of different organs and is compatible with high-throughput imaging, RNA-seq, and transport assays. Rajasekar et al. developed iFlowPlate^TM^, a microfluidic platform using gravity-driven flow to culture up to 128 independently perfused and vascularized colon organoids [32]. Ramadan and colleagues reported on a microfluidic chip for immune cell activation and cytokine profiling. The chip contains an array of parallel channels that hold magnetic beads linked with antibodies specifically targeting the desired cytokine to allow direct cytokine profiling. [145].

In addition, using novel materials in gut-on-chip systems will help solve some of the current challenges, such as scalability, standardization, and miniaturization [146]. Nanomaterials are already being used in the development of biosensors [65], new drug carriers [147], and three-dimensional scaffolds supporting complex tissue cultures [148]. Moreover, some nanomaterials possess antimicrobial properties, which make them promising for microbiome research [149,150]. Additionally, the development of new hydrogels could solve the current variability and ethical issues associated with the use of Matrigel [151,152]. In summary, advances in biomaterials science are set to play a pivotal role in the development of new microfluidic systems for diagnostics and clinical applications.

Altogether these support the notion that organ-on-a-chip models could be employed for high-throughput drug screenings (HTS). However, HTS applications of on-chip models need to be scalable and standardized so that the performances of hundreds to thousands of chips used are consistent and robust. There are many challenges to achieving this, such as establishing a detailed pipeline that can include materials, automation, data management, and analysis. Protocols with time-sensitive steps, such as ECM polymerization or live dye imaging, may need a synchronized and coordinated workflow to ensure chip-to-chip consistency. Fast assays and readouts should be developed to acquire, handle, and process large volumes of data from many chips [140], as one sometimes needs to make quick decisions (within a few hours to a day). This is especially the case for quality control assays to check quality at various stages of the workflow so that one may exclude failed quality chips from exposure. Examples of quality control assays are live-dead assays to ascertain the viability of seeded cells, methods to exclude chips with poor morphologies or phenotypes (visual scoring, TEER), and rule-based algorithms to check the quality of data acquisition, analysis, and hit selection. Compatibility with industrial automated liquid handling and high throughput devices (plate reader, high content imagers) is preferred since they lead to lower cost barriers when adopting on-chip models for screens, or else customized equipment needs to be developed with additional investments. While standardizing the models, variations in the controls must be understood and isolated. If chips are in a classical multi-well format, common well-plate artifacts such as edge effects and compound cross-over need to be checked [153]. Various aspects, such as technical, biological, and clinical relevance, must be well-validated for reproductivity and repeatability (e.g., Luminex/ELISA performance, RNA-seq). Using guidelines established by previous cell-based assays, the correlation of variation (defined as 100% × standard deviation/mean) of the baseline control should be less than 20% [154], while the Z’ factor between the positive and negative controls should be at least 0 to make the model a screenable one [155].

## 6. Conclusions

Gut dysbiosis has been implicated in numerous diseases, but until recently, the tools to understand its underlying causes were limited. However, the advent of in vitro models, including gut-on-chip technology, has transformed the way we study intestinal physiology and host–microbial interactions. These models offer a more physiologically relevant approach, incorporating key features not found in traditional systems.

The use of gut-on-chip models in drug screenings can significantly reduce the reliance on animal research and pave the way for developing patient-specific, precision medicine treatments. In pre-clinical studies, these models can be employed to test the efficacy of potential therapies in a more accurate representation of human physiology, improving the predictiveness of pre-clinical studies for human clinical trials. In the clinical realm, gut-on-chip models can accelerate the development of new treatments by providing a controlled and standardized platform for studying disease mechanisms, evaluating the safety and efficacy of potential therapies, and monitoring treatment outcomes over time. They can also be used to personalize treatments based on an individual’s gut microbiome composition.

This review summarizes the key parameters of gut-on-chip models for studying host-microbial interactions. A comprehensive comparison of these systems is presented in Table 1. The selection of the appropriate model will depend on the specific research question, as each model has its advantages and disadvantages. We also discuss the future challenges the field has to address, such as standardization and scalability. In conclusion, gut-on-chip models have the potential to revolutionize the way we study host–microbial interactions and advance our understanding of disease mechanisms, ultimately leading to more effective and personalized treatments.

## Figures and Tables

**Figure 1 biomedicines-11-00619-f001:**
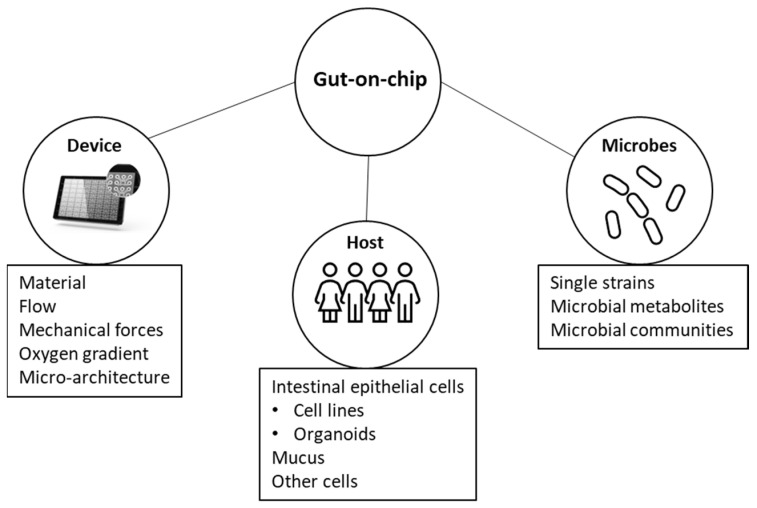
Features of gut-on-chip models to study host-microbial interactions.

**Table 1 biomedicines-11-00619-t001:** Gut-on-chip models to study host–microbial interactions reported in the literature.

Reference	Device Material	Membrane (Pore Size)	Intestinal Cells	Other Cells	Micro-Organisms	Co-Culture Duration	Anaerobic	Peristalsis	Micro-Architecture	Direct Contact of IECs and Microbes	Flow (Shear Stress)	Main Observation
Kim et al., 2012 [25]	PDMS	Yes (10 μM)	Caco-2	No	LGG	>1 week	No	Cyclic strain (10%, 0.15 Hz)	No	Yes	30 μL/h (0.02 dyne/cm^2^)	Observed that liquid flow and peristalsis induce spontaneous differentiation of Caco-2 cells into villi-like structures, polarization of the epithelium, and formation of barrier integrity. Successfully co-cultured LGG for more than a week.
Kim et al., 2016 [57]	PDMS	Yes (10 μM)	Caco-2	PBMCs	EIEC, VSL#3, *E. coli*	>1 week	No	Cyclic strain (10%, 0.15 Hz)	No	Yes	30 μL/h (0.02 dyne/cm^2^)	Established a stable co-culture with pathogenic and commensal microbes. Observed that cessation of peristalsis-like motions induced bacterial overgrowth. Recapitulated intestinal infection and inflammatory responses.
Shah et al., 2016 [21]	Polycarbonate (PC)	Yes (1 μM, 50 nm)	Caco-2	CD4+ T cells	LGG, *B. caccae*	24 h	Yes	No	No	No	1500 μL/h (not specified)	Developed a modular platform allowing real-time oxygen monitoring and the transcriptome analysis of the co-culture.
Villenave et al., 2017 [75]	PDMS	Yes (10 μM)	Caco-2	No	CVB1	24 h	No	Cyclic strain (10%, 0.15 Hz)	No	Yes	30 μL/h (0.02 dyne/cm^2^)	Showed successful human enterovirus infection in a gut-on-chip model.
Shin et al., 2018 [125]	PDMS	Yes (10 μM)	Caco-2	PBMCs	*E. coli*, VSL#3, LPS	24 h	No	Cyclic strain (10%, 0.15 Hz)	No	Yes	50 μL/h (not specified)	Studied probiotic administration before and after epithelial injury. Probiotic administration after injury exacerbated the infection, while probiotic administration before injury prevented the secretion of inflammatory cytokines and promoted the secretion of mucus.
Tovaglieri et al., 2019 [29]	PDMS	Yes (7 μM)	Patient-derived organoids	HIMECs	EHEC	6 h	No	No	No	Yes	60 μL/h (not specified)	Recapitulated EHEC infection in vitro and species differences in sensitivity to this pathogen. Identified four metabolites promoting EHEC infection in humans.
Grassart et al., 2019 [58]	PDMS	Yes (not reported)	Caco-2	No	*Shigella flexneri*	2 h	No	Cyclic strain (10%, 0.15 Hz)	No	Yes	30 μL/h (0.0009 dyne/cm^2^)	Replicated the hallmarks of *Shigella* infection. Observed that Shigella exploits gut architecture and mechanical forces to maximize infectivity.
Shin et al., 2019 [64]	PDMS	Yes (10 μM)	Caco-2	No	2 obligates anaerobes	72 h	Yes	Cyclic strain (10%, 0.15 Hz)	No	Yes	50 μL/h (0.02 dyne/cm^2^)	Created an anoxic–oxic interface-on-a-chip with a controlled oxygen gradient allowing the culture of two obligate anaerobes for up to a week.
Jalili-Firoozinezhad et al., 2019 [67]	PDMS	Yes (not reported)	Caco-2, Patient-derived organoids	HIMECs	complex microbiota (200 taxonomic units)	5 days	Yes	Cyclic strain (10%, 0.15 Hz)	No	Yes	60 μL/h (not specified)	Co-cultured Caco-2 or primary IECs in direct contact with complex microbiota (200 species) over 5 days in an anaerobic intestine-on-chip.
Maurer et al., 2019 [20]	Polystyrol (PS)	Yes (8 μM)	Caco-2	HUVECs, PMBC-derived macrophages, and dendritic cells	LGG, *C. albicans*, LPS	24 h	No	No	No	Yes	Endothelial side: 3000 μL/h (0.7 dyne/cm^2^), Luminal side: 3000 μL/h (0.1 dyne/cm^2^)	Developed a co-culture of PBMCs, Caco-2, and HUVEC displaying physiological immune tolerance of the lumen to microbial PAMPs. Observed that infection with LGG reduces *C. albicans* tissue damage, fungal burden, and yeast translocation.
Sunuwar et al., 2020 [60]	PDMS	Yes (7 μM)	Patient-derived organoids	No	*E. coli* HS toxin	NA	No	Cyclic strain (10%, 0.15 Hz)	No	NA	60 μL/h (not specified)	Demonstrated that flow—but not peristalsis—affects the response to *E. coli* heat-stable enterotoxin in human jejunal organoids.
Guo et al., 2021 [132]	PDMS	Yes (5 μM)	Caco-2, HT-29	HUVECs, PBMCs	SARS-CoV-2	3 days post-infection	No	No	No	Yes	200 μL/h apical, 50 μL/h basal (not specified)	Created an intestinal SARS-CoV-2 infection model that recapitulated relevant intestinal pathophysiology.
Jing et al., 2020 [28]	PDMS	Yes (10 μM)	Caco-2	HUVECs, human macrophages U937	*L. casei, E. coli*	4 days	No	Cyclic strain (10%, 0.15 Hz)	No	Yes	60 μL/h (not specified)	Observed that periodic peristalsis, fluid flow, and the presence of HUVECs promoted the proliferation and differentiation of Caco-2. Co-cultured *L. casei* with Caco-2 cells for up to a week and observed protection against *E. coli* overgrowth. Administered antibiotics, which suppressed *E. coli*-induced intestinal inflammation and injury.
Nikolaev et al., 2020 [42]	PDMS	NA (single channel)	Mouse organoids	No	*C. parvum*	20 days	No	No	Yes, 50–75 μM wide, 170 μM long	Yes	Not specified	Developed gut-on-a-chip models recapitulating the spatial arrangement of the crypt-villus axis, containing rare, specialized cells, and remaining stable for several days in culture, allowing long-term host-microbial studies.
Shin et al., 2020 [66]	PDMS	Yes (not reported)	Caco-2, Patient-derived organoids	No	Fecal microbiome	2 days	No	Cyclic strain (5%, 0.15 Hz)	Convoluted channel	Yes	50 μL/h (not specified)	Developed a system with a convoluted channel and multiaxial deformations. Observed that patient-derived organoids formed epithelial layers with disease-specific differentiations (UC, CD, CR). Co-cultured microbiome in an anoxic–oxic interface and observed the formation of microcolonies.
Yuan et al., 2020 [130]	PDMS	Yes (0.4 μm)	Caco-2	No	*B. breve*, *E. coli Hu734*	96 h (*B. breve*), 48 h (*E. coli*)	No	No	No	Yes	30 μL/h (not specified)	Observed protective effects of *B. breve* against *E. coli* with the measurement of layer thickness and membrane surface coverage by IECs.
Gazzaniga et al., 2021 [117]	PDMS	Yes (7 μM)	Mouse organoids	No	*S. typhimurium*, *E. faecium*, human microbiome stock, mouse microbiome stock	16 h	Yes	No	No	Yes	Not specified	Developed a mouse colon chip to study *S. typhimirium* infection and symbiosis between microbiome bacteria and intestinal epithelium.
Nelson et al., 2021 [127]	PDMS	Yes (7 μM)	Caco-2, HT-29	HMVECs	LBP SYN5183	12 h	No	No	No	Yes	60 μL/h (not specified)	Synthetic live bacterial therapeutic to treat phenylketonuria in human gut-on-a-chip.
Jeon et al., 2022 [36]	PDMS	No	Caco-2	HUVECs	*L. Plantarum* probiotics (HY7715 and ATCC14917), *B. lactis* probiotic (HY8002)	5 days	No	No	No	Yes	Not specified	Observed that the co-culture of probiotics with a damaged epithelial layer resulted in the recovery of barrier function without bacterial overgrowth.
Chin et al., 2022 [133]	PDMS	NA, single channel	HT-29	NA	*E. coli*, T4 phages	24 h	No	No	No	Yes	120 μL/h (0.025 dyne/cm^2^)	Successfully co-cultured phages and bacteria in a human gut-on-chip model.
Jing et al., 2022 [126]	PDMS	Yes (10 μM)	Caco-2	HUVECs, primary macrophages	*E. coli* 11775	4 days	No	Cyclic strain (15%, 0.15 Hz)	No	Yes	60 μL/h (not specified)	Observed that chitosan oligosaccharides can protect against intestinal and vascular damage from *E. coli* 11775.
De Gregorio et al., 2022 [68]	PDMS	No	Caco-2	PBMCs, hISEMFs	*L. rhamnosus*, *B. longum*	16 h	Yes	No	No	Yes	1800 μL/h (0.0267 dyne/cm^2^)	Developed an immuno-competent gut-microbiota axis with a complex serosal environment consisting of a responsive ECM and immune mediator released from different cell types (epithelial, stromal, blood, microbes).
Boquet-Pujadas et al., 2022 [59]	PDMS	Yes (7 μM)	Caco-2	NA	*E. histolytica*, *S. flexneri*	2 h (*S. flexneri*), 7 h (*E. histolytica*)	No	Cyclic strain (10%, 0.15 Hz)	No	Yes	30 μL/h (not specified)	Observed that peristalsis is a determinant for the invasion of *E. histolytica* and *S. flexneri* although they have different mechanisms of infection.
Min et al., 2022 [131]	PDMS	Yes (10 μM)	Caco-2	NA	LGG, VSL#3	72 h	No	Cyclic strain (10%, 0.15 Hz)	No	Yes	50 μL/h (~0.003 dyne/cm^2^)	Administered live probiotic bacteria in a leaky intestinal epithelium and observed the recovery of barrier function and mucosal inflammation.
Zhao et al., 2022 [27]	PDMS	Yes (8 μM)	Caco-2	HUVECs, PBMCs	LGG, *E. coli* (ESBL-EC)	24 h	No	No	No	Yes	60 μL/h (not specified)	Studied the effects of LGG and antibiotics against drug resistant bacteria and observed that Amikacin efficiently inhibits ESBL-EC-induced inflammation, as observed in clinical studies.

PDMS: polydimethylsiloxane; LGG: *lactobacillus rhamnosus* GG; HUVECs: human umbilical vein endothelial cells; PBMCs: human peripheral blood mononuclear cells; hISEMFs: human intestinal subepithelial myofibroblasts; HMVECs: human microvascular endothelial cells; HIMECs: human intestinal microvascular endothelial cells; EHEC: enteroinvasive *E. coli*; CVB1: Coxsackievirus B1; IECs: intestinal epithelial cell; ECM: extracellular matrix.

## Data Availability

Not applicable.

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
