# Peer review of "Gut-on-a-Chip Models: Current and Future Perspectives for HostMicrobial Interactions Research"

_biomedicines, 2023, doi:10.3390/biomedicines11020619_

Round 1

Reviewer 1 Report

There are other review articles published recently in similar topic areas (example: The Progress of Intestinal Epithelial Models from Cell Lines to Gut-On-Chip. Int. J. Mol. Sci. 2021, 22, 13472. https://doi.org/10.3390/ijms222413472).Please review other review articles and describe in the text how is yours different. What new are you adding to existing review work?

Please check for plagiarism. There are parts of sentences that are from other journal papers. Please paraphrase and cite sources used. Please make sure you cite all work that you used in this review. There are several sections that require citation.

Please pay attention to repetition of the same words and rephrase accordingly. For example, there is repetition of the word "Intestinal": "Intestinal organoids have emerged as a new promising tool to model intestinal physiology and host-microbial interactions because they contain numerous specific intestinal cell subtypes and can recapitulate regional identity". Frequent use of the word "recapitulate/recapitulation".

In line 426, De Gregorio et al proposed or developed ? De Gregorio had coauthors (line 432 "the author observed"). Please improve this paragraph.

In line 453: "Lastly, the combination of gut-on-chip with other systems such as bioreactors could bring new solutions to cover some of the drawbacks mentioned [133,134]." What are these drawbacks?

Reviewer 2 Report

This manuscript deals with " Gut-on-a-chip models: current and future perspectives for host-microbial interactions research". This article claims that using of  Gut-on-a-chip models could be a suitable for understanding of  disease mechanisms and the  as well as effect of microbes on the gut. The topic is promising and i really enjoyed. Therefore, I suggest a minor correction and require a detailed clarification.

Correction to be addressed by the authors as follows: The abstract is not well organized, where the sentences are incomplete and no continuity is there. It would be feasible, if include the significance of the current study in the abstract. A brief description of how the authors selected information from the literature in the databases, as well as what time period they searched for, is missing. Authors should justify and expand the information on the diseases in which this species is mentioned, highlighting the main contribution in the preclinical and clinical fields.

The major text of this manuscript is focused to some mechanisms but not in the case of pharmacological activities and clinical applications. Authors should specify the main experimental conditions used on the evidences from the literature. Where they briefly describe the most important data reported in the literature in a homogeneous manner and sequence reinforcing the relevance of this species as medicinal alternative. Authors should discuss whether the use of novel materials including nanomaterials and biomaterials on chip models represents a solid alternative to existing diagnostic methods. Please add more previous studies to your manuscript in discussion section using these papers: -DOI:10.1016/j.jksus.2021.101710  -DOI:10.1016/j.jcis.2020.10.047   -DOI:10.3389/fbioe.2022.855136

Conclusions should reaffirm the fundamental contribution of this Review.

Round 2

Reviewer 1 Report

Most of the concerns are addressed. However, still frequent use of the word "recapitulate/recapitulation". Please review for English.